# Reaction between Criegee intermediates and hydroxyacetonitrile: Reaction mechanisms, kinetics, and atmospheric implications

Chaolu Xie<sup>1</sup> Shunyu Li<sup>2</sup> Bo Long\*1,2

10

Correspondence to: Bo Long (wwwltcommon@sina.com)

Abstract. Hydroxyacetonitrile (HOCH<sub>2</sub>CN) is released from wildfires and bleach cleaning environments, which is harmful to the environment and human health. However, its atmospheric lifetime remains unclear. Here, we theoretically investigate the reactions of Criegee intermediates (CH<sub>2</sub>OO and *syn*-CH<sub>3</sub>CHOO) with HOCH<sub>2</sub>CN to explore their reaction mechanisms and obtain their quantitative kinetics. Specifically, we employ computational strategies approaching CCSDT(Q)/CBS accuracy, combined with a dual-level strategy, to unravel the key factors governing the reaction kinetics. We find an unprecedentedly low enthalpy of activation of −5.61 kcal/mol at 0 K for CH<sub>2</sub>OO + HOCH<sub>2</sub>CN among CH<sub>2</sub>OO reaction with atmospheric species containing a C≡N group. Furthermore, we also find that the low enthalpy of activation is caused by hydrogen bonding interactions. Moreover, the present findings reveal the rate constant of CH<sub>2</sub>OO + HOCH<sub>2</sub>CN determined by loose and tight transition states has a significantly negative temperature dependence, reaching 10<sup>-10</sup> cm³ molecule<sup>-1</sup> s<sup>-1</sup> close to the collisional limit at below 220 K. In addition, our findings also reveal that the rate constant of CH<sub>2</sub>OO + HOCH<sub>2</sub>CN is 10³-10² times faster than that of OH + HOCH<sub>2</sub>CN below 260 K. The calculated kinetics in combination with data based on global atmospheric chemical transport model suggest that the CH<sub>2</sub>OO + HOCH<sub>2</sub>CN reaction dominates over the sink of HOCH<sub>2</sub>CN at southeast China, northern India at 1 km and in the Indonesian and Malaysian regions at 5 and 10 km. The present findings also reveal that the CH<sub>2</sub>OO + HOCH<sub>2</sub>CN reaction leads to the formation of glycolamide, which could contribution to the formation of secondary organic aerosols.

<sup>&</sup>lt;sup>1</sup> College of Physics and Mechatronic Engineering, Guizhou Minzu University, Guiyang 550025, China.

<sup>&</sup>lt;sup>2</sup>College of Materials Science and Engineering, Guizhou Minzu University, Guiyang 550025, China

<sup>\*</sup>corresponding author, E-mail: wwwltcommon@sina.com (Bo Long)

# 1 Introduction




Hydroxyacetonitrile (HOCH<sub>2</sub>CN), a reactive nitrogen-containing compound, has recently been identified as a C<sub>2</sub>H<sub>3</sub>NO isomer. Earlier field measurements had attributed the C<sub>2</sub>H<sub>3</sub>NO signal to methyl isocyanate (CH<sub>3</sub>NCO) when using chemical ionization mass spectrometry (CIMS) (Priestley et al., 2018; Mattila et al., 2020; Wang et al., 2022a), as CIMS is insensitive to the detection of isomers, and thus cannot differentiate between the isomers of CH<sub>3</sub>NCO and HOCH<sub>2</sub>CN. However, very recent detection with I<sup>-</sup> chemical ionization mass spectrometry (I-CIMS) identified the C<sub>2</sub>H<sub>3</sub>NO signal as HOCH<sub>2</sub>CN (Finewax et al., 2024). In other words, the CH<sub>3</sub>NCO detected in the atmosphere is essentially HOCH<sub>2</sub>CN. Therefore, the atmospheric sources of CH<sub>3</sub>NCO from previous investigations are actually the sources of HOCH<sub>2</sub>CN. Consequently, HOCH<sub>2</sub>CN is emitted in chemicals released from biomass burning, such as wildfires and agricultural fires, as well as in bleach cleaning environments (Mattila et al., 2020; Priestley et al., 2018; Wang et al., 2022a; Koss et al., 2018; Papanastasiou et al., 2020).

Previous studies have demonstrated that HOCH<sub>2</sub>CN is secondary pollutant with negative impacts on the environment and human health (Worthy, 1985; Etz et al., 2024; Zhang et al., 2025). Specifically, smoke with HOCH<sub>2</sub>CN can be injected into the stratosphere through pyrocumulonimbus clouds, altering the composition of stratospheric aerosols, depleting the ozone layer, and affecting the Earth's radiation balance (Bernath et al., 2022; Ma et al.; Katich et al., 2023). Additionally, HOCH<sub>2</sub>CN is harmful to human health, including damage to the respiratory system and skin (Ganguly et al., 2017; Bucher, 1987). Therefore, understanding the chemical processes of HOCH<sub>2</sub>CN is important in the atmosphere.

The atmospheric lifetimes of HOCH<sub>2</sub>CN in the gas phase are not well understood. Hydroxyl radical (OH) is the most prevalent oxidant in the atmosphere (Wang et al., 2021). The generally considered removal for this species is through the reaction with hydroxyl radical (OH). However, the rate constant of OH + HOCH<sub>2</sub>CN is very slow, about 2.6 × 10<sup>-13</sup> cm<sup>3</sup> molecule<sup>-1</sup> s<sup>-1</sup> at 298 K (Marshall and Burkholder, 2024). This leads to that OH makes limited contribution to the sinks of HOCH<sub>2</sub>CN in the atmosphere. Therefore, it is necessary to explore the other removal routes for HOCH<sub>2</sub>CN in the atmosphere.

Criegee intermediates are crucial compounds, resulting from the ozonolysis of unsaturated compounds in the atmosphere (Criegee, 1975; Osborn and Taatjes, 2015; Chhantyal-Pun et al., 2020a; Bunnelle, 1991; Chhantyal-Pun et al., 2020b). They play key roles in the chemical processes of atmosphere because they significantly contribute to the formation of hydroxyl radical (OH) and sulfuric acid during the nighttime (Novelli et al., 2014; Lester and Klippenstein, 2018; Kroll et al., 2002; Newland et al., 2018; Kukui et al., 2021). Additionally, they can form secondary organic aerosols via the formation of low-volatile organic compound percussors (Khan et al., 2018; Inomata et al., 2014; Docherty et al., 2005; Chhantyal-Pun et al., 2018). In particular, Criegee intermediates can initiate atmospheric reactions, resulting in additional sinks for atmospheric species such as the reactions of Criegee intermediates with formic acid, nitric acid, hydrochloric acid, and formaldehyde and so on (Khan et al., 2018; Peltola et al., 2020; Long et al., 2009; Chung et al., 2019; Foreman et al.,

2016; Luo et al., 2023). While reaction kinetics of Criegee intermediates with HOCH<sub>2</sub>CN are prerequisite for elucidating its chemical processes and finding new sink pathways in the atmosphere, its kinetics are unknown.

In this article, we have investigated the reactions of Criegee intermediates (CH<sub>2</sub>OO and *syn*-CH<sub>3</sub>CHOO) with HOCH<sub>2</sub>CN by using specific computational strategies and methods to obtain quantitative enthalpies of activation at 0 K for R1 and R2 (See Scheme 1). Then, we used a dual-level strategy to obtain their quantitative rate constants under atmospheric conditions. In our dual-level strategy, W3X-L//DF-CCSD(T)-F12b/jun-cc-pVDZ is used to obtain conventional transition state theory rate constants, while validated DFT methods capture recrossing and tunnelling effects through CVT with small-curvature tunnelling. Additionally, torsional anharmonicity and anharmonicity are considered in kinetics calculations. We also considered the decomposition process of the intermediate product formed in R1. Finally, we discuss the importance of these reactions investigated here by comparing with the corresponding OH radical reactions combined with the atmospheric concentrations of these species based on global atmospheric chemistry transport model GEOS-Chem (http://www.geoschem.org).

$$HOCH_2CN + CH_2OO$$

$$\xrightarrow{TS1} HO$$
 $C$ 
 $CH_2$ 
 $CH_2$ 

$$HOCH_2CN + syn\text{-}CH_3CHOO \xrightarrow{\text{s-TS1}} HO \xrightarrow{\text{C}} CH_3 \text{ (R2)}$$

Scheme 1. The reaction route for CH<sub>2</sub>OO/syn-CH<sub>3</sub>CHOO) + HOCH<sub>2</sub>CN

# 2 Computational methods and strategies





# 2.1 Electronic structure methods and strategies

For simplicity, the activation enthalpy at 0 K is defined as the difference between the transition state and the reactants, abbreviated as  $\Delta H_0^{\ddagger}$ . The difference between the products and the reactants is defined the reaction enthalpy at 0 K, abbreviated as  $\Delta H_0$ .

Morden quantum chemical methods and reaction rate theory can be used to obtain quantitative kinetics for atmosphere reactions (Long et al., 2018). However, the calculated processes are very complex, where error bars are controlled by multiple parameters. Furthermore, the parameters are correlated with each other. Gas-phase chemical reactions also crucially depend on entropies in thermalized conditions at non-zero temperatures, as well as excess energy, collisional stabilization, and fall-off effects in non-thermalized conditions. However, accurate determination of  $\Delta H_0^{\ddagger}$  remains essential for calculating

quantitative kinetics. The quantitative  $\Delta H_0^{\ddagger}$  that is determined by optimized geometries, zero-point vibrational energies, and single point energies in electronic structure methods (Long et al., 2019a).





Our previous investigations have shown that W3X-L (Chan and Radom, 2015)//DF-CCSD(T)-F12b/jun-cc-pVDZ (Győrffy and Werner, 2018; Parker et al., 2014) can be utilized to obtain quantitative ΔH<sub>0</sub><sup>‡</sup> for the bimolecular reactions containing Criegee intermediate (Wang et al., 2022b; Long et al., 2021; Xie et al., 2024; Zhang et al., 2024). Here, we used W3X-L//DF-CCSD(T)-F12b/jun-cc-pVDZ to investigate the CH<sub>2</sub>OO + HOCH<sub>2</sub>CN reaction. Additionally, CCSD(T)-F12a/cc-pVDZ-F12 was used to validate the reliability of DF-CCSD(T)-F12b/jun-cc-pVDZ for optimized geometries and calculated frequencies in the CH<sub>2</sub>OO + HOCH<sub>2</sub>CN reaction. Thus, with respect to *syn*-CH<sub>3</sub>CHOO + HOCH<sub>2</sub>CN, the validated DF-CCSD(T)-F12b/jun-cc-pVDZ method was used to do geometrical optimization and frequency calculations. In single point energy calculations, it is noted that W3X-L is equal to W2X and post-CCSD(T) components (Chan and Radom, 2015). Here, we assume that post-CCSD(T) contribution of *syn*-CH<sub>3</sub>CHOO + HOCH<sub>2</sub>CN approximately equates to the contribution from CH<sub>2</sub>OO + HOCH<sub>2</sub>CN. We used the approximation strategy mentioned previously (Sun et al., 2023) to obtain the accuracy close to W3X-L in equation (1) for *syn*-CH<sub>3</sub>CHOO + HOCH<sub>2</sub>CN to reduce computational costs.

$$\Delta E_{\text{BE}}^{\ddagger,s-\text{TS1}} = \Delta E_{\text{W2X}}^{\ddagger,s-\text{TS1}} + \Delta E_{\text{W3X-L-W2X}}^{\ddagger,\text{TS1}} \tag{1}$$

Here,  $\Delta E_{\rm BE}^{\pm,s-{\rm TS1}}$  is the barrier height for the transition state s-TS1 (See Scheme 1).  $\Delta E_{\rm W2X}^{\pm,s-{\rm TS1}}$  is the barrier height for s-TS1 calculated by W2X.  $\Delta E_{\rm W3X-L-W2X}^{\pm,{\rm TS1}}$  is post-CCSD(T) contribution that comes from the difference between W3X-L and W2X in TS1 (See Fig. 1 and 2). The benchmark methods are called higher level (HL) structure methods; this helps to clearly illustrate the dual-level strategy for kinetics calculations discussed below.

A reliable density functional method was chosen in comparison to the benchmark results to perform direct kinetics calculations. Here, we chose M06-CR (Long et al., 2016)/MG3S (Zhao et al., 2005) and M11-L (Peverati and Truhlar, 2012) /MG3S functional method for the reactions of CH<sub>2</sub>OO and *syn*-CH<sub>3</sub>CHOO with HOCH<sub>2</sub>CN due to the mean unsigned error (MUD) of 0.23 kcal/mol and 0.72 kcal/mol as listed in Table 1, respectively. M06CR/MG3S and M11-L/MG3S are called lower level (LL) electronic structure method in the present work. The intrinsic reaction coordinate (IRC) was performed by M11-L/MG3S, and the results were depicted the results in Figure S3.

Table 1. The enthalpies of activation at 0 K for the transition states of the CH<sub>2</sub>OO/syn-CH<sub>3</sub>CHOO + HOCH<sub>2</sub>CN reactions by various theoretical methods (in kcal/mol).

| Methods                                   | $\Delta H_0^{\ddagger}$ TS1 | MUD  |  |
|-------------------------------------------|-----------------------------|------|--|
| CH <sub>2</sub> OO + HOCH <sub>2</sub> CN | 151                         |      |  |
| W3X-L//DF-CCSD(T)-F12b/jun-cc-pVDZ        | -5.61                       | 0.00 |  |
| M06CR/MG3S                                | -5.38                       | 0.23 |  |
| W2X//CCSD(T)-F12a/cc-pVDZ-F12             | -6.16                       | 0.55 |  |
| W2X//DF-CCSD(T)-F12b/jun-cc-pVDZ          | -6.19                       | 0.58 |  |
| CCSD(T)-F12a/cc-pVDZ-F12                  | -5.98                       | 0.37 |  |
| DF-CCSD(T)-F12b/jun-cc-pVDZ               | -6.77                       | 1.16 |  |
| M11-L/MG3S                                | -7.52                       | 1.91 |  |

| syn-CH <sub>3</sub> CHOO + HOCH <sub>2</sub> CN              |       |      |
|--------------------------------------------------------------|-------|------|
| $\Delta E_{BE}^{\dagger,s-TS1}//DF-CCSD(T)-F12b/jun-cc-pVDZ$ | -1.39 | 0.00 |
| W2X//DF-CCSD(T)-F12b/jun-cc-pVDZ                             | -1.97 | 0.58 |
| M11-L/MG3S                                                   | -2.11 | 0.72 |
| DF-CCSD(T)-F12b/jun-cc-pVDZ                                  | -3.13 | 1.74 |
| M06-CR/MG3S                                                  | 0.47  | 1.86 |

# 105 2.2 Scale factors for calculated frequencies

Previous studies have verified that the standard scale (See Table S1) is suitable for reactants and some transition states (Bao et al., 2016b; Zhang et al., 2017). In order to further explore the effect of anharmonicity on the zero-point vibrational energy, the calculation of the specific reaction scale factors was carried out. The results in Tables S2 and S4 show that the anharmonicity can be neglected in calculating  $\Delta H_0^{\ddagger}$ . Details of the calculations can be found in previous work of Long et al (Long et al., 2023). Therefore, we used standard scale factor in this work.

# 2.3 Kinetics methods





The rate constants of the reaction (R1) are calculated by considering the tight and loose transition states because of its low-temperature close-to-collision-limit rate constant. Here, the loose transition state refers to the process from the reactants to the pre-reaction complexes, while the tight transition state is the process from the reactants to the products via the transition state TS1 in Figure 1. The steady state approximation based on the unified statistical theory (CUS) (Garrett and Truhlar, 1982; Bao and Truhlar, 2017; Zhang et al., 2020; Long et al., 2024) is used to calculate the total rate constant by simultaneously considering both transition states by

$$k_{CUS}(T) = \frac{k_{tight}k_{loose}}{k_{tight} + k_{loose}} \tag{1}$$

where k<sub>tight</sub> was calculated by using a dual-level strategy described in detail below, while k<sub>loose</sub> was calculated by variable-reaction-coordinate variational transition state theory (VRC-TST) (Zheng et al., 2008; Bao et al., 2016a; Georgievskii and Klippenstein, 2003). In VRC-TST calculation, the reaction coordinate *s* is obtained by defining the distance between one pivot point on one reactant and the other pivot point on the other, and the dividing surface is defined by the pivot point connected to each reactant. The pivot point is located in a vector at a distance d from the centre of mass (COM) of the reactants, which is chosen to minimize the reaction rate. The vector connecting the pivot point to the centre of mass of reactant and is perpendicular to the plane of the reactant. The distance *s* between the pivot points was varied between 2.6 and 10 Å in steps of 0.1 Å to find the optimum value. Simultaneously, 500 Monte Carlo sampling points were used to sample single-faceted dividing surfaces. The VRC-TST calculation were performed by minimizing the rate constant by changing the distance between two pivot points and the location of the pivot points. The VRC-TST were performed using M06-CR/MG3S

for reaction R1. However, the high-pressure-limited rate constants of the reaction (R2) were calculated only by considering tight transition state s-TS1.

The dual-level strategy has been put forward and used in previous works (Long et al., 2022; Long et al., 2018, 2019b; Xia et al., 2022; Gao et al., 2024; Long et al., 2016; Sun et al., 2023). The strategy combines the theory of conventional transition states (Glasstone et al., 1941) on the HL with the theory of canonical variational transition states (Garrett and Truhlar, 1979; Truhlar et al., 1982) on the LL and takes into account the small curvature tunnelling effect (Liu et al., 1993). In addition, the torsional anharmonicity factor is considered in our strategy (Long et al., 2023; Zhang et al., 2024; Sun et al., 2023; Li and Long, 2024; Xie et al., 2024; Jiang et al., 2025). The rate constant is given by eqn (2),

$$k = k(T)_{\text{HL}}^{\text{TST}} \kappa_{LL}^{\text{SCT}}(T) \Gamma_{LL}(T) \Gamma_{fwd,LL}^{\text{MS-T}}$$
(2)

where  $k(T)_{\rm HL}^{\rm TST}$  is the rate constant without recrossing and tunneling effects calculated at HL.  $\kappa_{LL}^{SCT}(T)$  and  $\Gamma_{LL}(T)$  is referred to tunneling transmission coefficient and recrossing transmission coefficient calculated by LL, respectively.  $F_{fwd,LL}^{\rm MS-T}$  is torsional anharmonicity factor calculated by using multi-structural method with coupled torsional potential and delocalized torsions (MS-T(CD) method) (Zheng and Truhlar, 2013; Chen et al., 2022). Three factors were calculated at the validated density functional methods at LL. The calculation each component present in equation (2) were provided in Tables S5 and S6.

# 2.4 Software



All density functional calculations were performed by using Gaussian 16 (Frisch et al., 2016) and MN-GFM (Zhao et al., 2015) for geometry optimization and frequency calculations and all coupled cluster calculations by using Molpro 2022 (Werner et al., 2012) and MRCC code (Kállay et al., 2020). Direct kinetics calculations were performed using Polyrate 2017 (Zheng et al., 2017b) and Gaussrate 2017-C (Zheng et al., 2017a). Torsional anharmonicity factor was calculated by using MSTor 2022 code (Zheng et al., 2012). And rate constants were calculated by Kisthelp program package (Canneaux et al., 2014).

# 3 Results and Discussion

#### 3.1 Electronic Structure calculation results

# 3.1.1 The reaction of CH<sub>2</sub>OO + HOCH<sub>2</sub>CN

The reaction of CH<sub>2</sub>OO + HOCH<sub>2</sub>CN has not been reported in the literature. It is noted that there are three different functional groups (H-O, C≡N, and CH<sub>2</sub>) in HOCH<sub>2</sub>CN. Therefore, we explored four different mechanisms of CH<sub>2</sub>OO + HOCH<sub>2</sub>CN as described in Fig. 1. Three of them are similar to the CH<sub>2</sub>OO + CH<sub>3</sub>CN reaction, as they contain the same C-H

Figure 1: The relative enthalpies at 0 K for the reaction of CH<sub>2</sub>OO + HOCH<sub>2</sub>CN. Values are given for all species as calculated by M11-L/MG3S.

The most feasible (first) mechanism leads to the formation of five-membered cyclic intermediate M1 via  $C \equiv N$  group addition to COO group in Fig. 1. Specifically, carbon atom of  $C \equiv N$  group in HOCH<sub>2</sub>CN is added to the terminal oxygen atom of CH<sub>2</sub>OO and N atom of C $\equiv N$  group in HOCH<sub>2</sub>CN is added to the central carbon of CH<sub>2</sub>OO by the transition state TS1 (See Fig. 1). This mechanism is the same as CH<sub>2</sub>OO + CH<sub>3</sub>CN and is like that of the reaction between CH<sub>2</sub>OO and carbonyl group (Long et al., 2021; Luo et al., 2023; Chhantyal-Pun et al., 2018; Chung et al., 2019). We called the


mechanism as carbon-oxygen addition coupled carbon-nitrogen addition mechanism. The second mechanism still occurs via five-membered cyclic transition states TS1a and TS1b responsible for the formation of M1a in Fig. 1. The H atom of the OH group in HOCH<sub>2</sub>CN is migrated to the terminal oxygen atom of CH<sub>2</sub>OO, and simultaneously the oxygen atom of OH group in HOCH<sub>2</sub>CN is added to the central carbon atom of CH<sub>2</sub>OO by TS1a and TS1b, which is similar to the reaction of CH<sub>2</sub>OO with molecules containing OH group such as  $H_2O/H_2O_2/HOOCH_2SCHO/CH_3C(O)OOH/HOCI$  (Long et al., 2016; Zhao et al., 2022; Long et al., 2024; Zhang et al., 2024; Xie et al., 2024). The third mechanism is the addition of C atom on the C=N group in HOCH<sub>2</sub>CN to the central C atom of CH<sub>2</sub>OO and the N atom of C=N group in HOCH<sub>2</sub>CN to the terminal O of CH<sub>2</sub>OO by TS1d. The last mechanism is that the H atom on the central CH<sub>2</sub> group in HOCH<sub>2</sub>CN shifts to the terminal O atom of CH<sub>2</sub>OO and the C atom is added to the central C atom of CH<sub>2</sub>OO via TS1c, which leads to the formation of a peroxide. We mainly consider the most feasible mechanism in detail in this work because  $\Delta H_0^{\ddagger}$  of TS1 is at least 5 kcal/mol lower than those of other reaction pathways by M11-L/MG3S (See Fig. 1). Moreover, the intermediate product M1 formed has a larger  $\Delta H_0$  of -58.33 kcal/mol in Fig.1. In addition, the calculated Gibbs free energy barriers at 298 K also show the five-ring closure reaction pathway via TS1 is the lowest in the CH<sub>2</sub>OO + HOCH<sub>2</sub>CN reaction (See Figure S2); this reveals that entropy has a negligible effect on reaction mechanism.

We previously showed that CCSD(T)-F12a/cc-pVDZ-F12 can reach the accuracy of CCSD(T)-F12a/cc-pVTZ-F12 for geometrical optimization and frequency calculations for reactions containing C≡N groups (Long et al., 2021; Zhang et al., 2024; Zhang et al., 2022). Therefore, we further show the reliability of DF-CCSD(T)-F12b/jun-cc-pVDZ by using CCSD(T)-F12a/cc-pVDZ-F12. As a result, the difference between W2X//CCSD(T)-F12a/cc-pVDZ-F12 and W2X//DF-CCSD(T)-F12b/jun-cc-pVDZ shows that DF-CCSD(T)-F12b/jun-cc-pVDZ is only 0.03 kcal/mol for ΔH<sub>0</sub><sup>‡</sup> of TS1 (See Table 1); this further shows that DF-CCSD(T)-F12b/jun-cc-pVDZ is quantitatively reliable for geometrical optimizations and frequency calculations in the preset investigations.

 $\Delta H_0^{\ddagger}$  of R1 via TS1 is computed to be -5.61 kcal/mol calculated by W3X-L//DF-CCSD(T)-F12b/jun-cc-pVDZ-F12, which is 5.25 lower than that of CH<sub>2</sub>OO + CH<sub>3</sub>CN (Zhang et al., 2022) calculated by W3X-L//CCSD(T)-F12a/cc-pVTZ-F12. The much lower  $\Delta H_0^{\ddagger}$  via TS1 in R1 leads to much faster rate constant of R1, comparing with the CH<sub>2</sub>OO + CH<sub>3</sub>CN reaction. Simultaneously, we also found that  $\Delta H_0^{\ddagger}$  of TS1 is 2.05 kcal/mol lower than that of the (CF<sub>3</sub>)<sub>2</sub>CFCN + CH<sub>2</sub>OO reaction calculated using the best estimate (Jiang et al., 2025). From geometrical point of view, the much lower  $\Delta H_0^{\ddagger}$  via TS1 is due to the introduction of HO group in HOCH<sub>2</sub>CN, comparing with CH<sub>3</sub>CN; this remarkably change the reactivity of HOCH<sub>2</sub>CN toward CH<sub>2</sub>OO. We note that the introduction of OH in HOCH<sub>2</sub>CN results in the formation of hydrogen bonding in TS1. The hydrogen bonding is formed via the interaction OH group in HOCH<sub>2</sub>CN with the terminal oxygen atom in CH<sub>2</sub>OO in TS1. The bond distance between the hydrogen atom of HO group in HOCH<sub>2</sub>CN and the terminal oxygen atom of CH<sub>2</sub>OO in TS1 is computed to be 1.942 Å by DF-CCSD(T)-F12b/jun-cc-pVDZ (See Fig. 2); this shows the formation of hydrogen bonding interaction opens a way for decreasing  $\Delta H_0^{\ddagger}$ .

Figure 2: The relative enthalpies at 0 K for the reaction of CH<sub>2</sub>OO/syn-CH<sub>3</sub>CHOO + HOCH<sub>2</sub>CN for R1 and R2. Values are given for all species as calculated by M11-L/MG3S, and in small parentheses and brackets, values are given for the transition state TS path as calculated by W3X-L//DF-CCSD(T)-F12b/jun-cc-pVDZ. The bond length in TS1 is in units of Å.

CCSD(T) has been considered as "gold standard" in the quantum chemical calculations. However, the previous investigations have shown that post-CCSD(T) is necessary to obtain quantitative reaction energy barriers for atmospheric reactions (Long et al., 2019b, 2016; Hansen et al., 2022; Xia et al., 2024). Here, we discuss the contribution of post-CCSD(T) calculations. The contribution of post-CCSD(T) is 0.58 kcal/mol from the difference between W3X-L and W2X of TS1, which is the same as the result of the reaction of CH<sub>2</sub>OO with CH<sub>3</sub>CN (0.58 kcal/mol) (Zhang et al., 2022) and our previous estimated value (0.50 kcal/mol) (Zhang et al., 2022; Zhao et al., 2022). This shows that post-CCSD(T) calculations are necessary for obtaining quantitative  $\Delta H_0^{\ddagger}$  for the reaction of Criegee intermediate with HOCH<sub>2</sub>CN. The MUD of M06-CR/MG3S is 0.23 kcal/mol for CH<sub>2</sub>OO and HOCH<sub>2</sub>CN in Table 1. Therefore, M06-CR/MG3S was chosen to perform direct kinetics calculation for the reaction of CH<sub>2</sub>OO with HOCH<sub>2</sub>CN.

Decomposition pathways for the formed product M1 have also been investigated at the M11-L/MG3S level; this is similar to the product decomposition pathway in the CH<sub>2</sub>OO + CH<sub>3</sub>CN reaction (Zhang et al., 2022). Firstly, the product M1 undergoes oxygen-oxygen bond cleavage via M1-TSa to result in forming a singlet biradical intermediate M2, which is similar to the reaction of CH<sub>2</sub>OO + HCHO (Jalan et al., 2013). Subsequently, the intermediate M2 undergoes the formation of carbon-nitrogen bonding to form a three-member ring with  $\Delta H_0^{\ddagger}$  of 21.76 kcal/mol relative to M1 via the transition state M2-TSb. Then, the three-member ring intermediate M3 then undergoes two different reaction routes. One is open-ring coupled hydrogen shift to form M4 via the transition state M3-TSc. The other is analogous to that proposed by Franzon et al. (Franzon et al., 2023), which is an open-ring coupled bond breaking to form HCHO and HOCH<sub>2</sub>NCO via the transition state M3-TSc1. However,  $\Delta H_0^{\ddagger}$  for M3-TS3c is 11.97 kcal/mol lower than that of M3-TSc1. Therefore, M3-TSc is the dominant reaction pathway for the unimolecular reaction of M3. Moreover, IRC calculations also show that M3-TSc connects well with M3 as described in Figure S3. Subsequently, the H atom of the intermediate OH on intermediate M4 is transferred to the N atom to yield the intermediate species M5. Then, the process was depicted in Fig. 3. The calculated enthalpy of reaction at 0 K of M5 is -86.55 kcal/mol, indicating the unimolecular isomerization is thermodynamically driven. Intermediate M5 undergoes unimolecular isomerization via two different pathways. In the first pathway, an intramolecular hydrogen transfer from the aldehyde group to the central carbonyl oxygen is followed by C-N bond cleavage, vielding CO and intermediate M6. Then, hydrogen shift of OH in M6 to NH group leads to the formation of glycolamide. Alternatively, a second pathway involves hydrogen migration from the aldehyde group to the central carbon atom, accompanied by C-N bond rupture, producing HNCO and glycolaldehyde. The formation of carbon monoxide proceeds with a significantly lower activation enthalpy (-54.19 kcal/mol) compared to that for glycolaldehyde (-32.32 kcal/mol), indicating that the COforming channel is kinetically favoured. Furthermore, the rate-determining step for the formation of the final product from the CH<sub>2</sub>OO + HOCH<sub>2</sub>CN reaction has been identified as the initial step, which is similar to the reaction of alkenes with ozone(Nguyen et al., 2015).






Figure 3: The relative enthalpies at 0 K for the decomposition reactions of the intermediate product M1 formed in the  $CH_2OO + HOCH_2CN$  reaction. Values are given for all species as calculated by M11-L/MG3S.

# 240 3.1.2 The reaction of syn-CH<sub>3</sub>CHOO + HOCH<sub>2</sub>CN



The reaction of syn-CH<sub>3</sub>CHOO with HOCH<sub>2</sub>CN has been also studied by considering similar mechanisms for the reaction of CH<sub>2</sub>OO + HOCH<sub>2</sub>CN. The HL calculation of  $\Delta H_0^{\ddagger}$  of s-TS1 was performed by employing an approximation method, as listed in eqn (1), which is discussed in method section. The lowest energy route is depicted in Fig. 2 and details can be found in Fig. S1.  $\Delta H_0^{\ddagger}$  of s-TS1 is -1.39 kcal/mol calculated by HL calculation, which is 4.22 kcal/mol higher than that of TS1. The lower reactivity of syn-CH<sub>3</sub>CHOO than CH<sub>2</sub>OO results in the much slower reaction of syn-CH<sub>3</sub>CHOO with HOCH<sub>2</sub>CN. However,  $\Delta H_0^{\ddagger}$  for s-TS1 is 5.45 kcal/mol is lower than the reaction of syn-CH<sub>3</sub>CHOO + CH<sub>3</sub>CN calculated by W3X-L//DF-CCSD(T)-F12b/jun-cc-pVDZ; this again shows that the introduction of OH group in HOCH<sub>2</sub>CN can significantly reduce  $\Delta H_0^{\ddagger}$  toward Criegee intermediates. However, the decrease in value of 5.45 kcal/mol between syn-CH<sub>3</sub>CHOO + HOCH<sub>2</sub>CN and syn-CH<sub>3</sub>CHOO + CH<sub>3</sub>CN; this indicates that the change in  $\Delta H_0^{\ddagger}$  is not only determined by the change from CH<sub>2</sub>OO to syn-CH<sub>3</sub>CHOO. We also found that the

MUD between the HL result and M11-L/MG3S is only 0.72 kcal/mol. Therefore, the combination of M11-L functional method with MG3S basis set was chose to perform direct kinetics calculations.

# 3.2 Kinetics






The rate constants for the reaction of R1 and R2 have been calculated and listed in Table 2. The details are listed in Table S5 and S6. We have fitted the calculated rate constants by eqn (3).

$$k_{\infty} = A \left( \frac{T + T_0}{300} \right)^n \exp \left[ -\frac{E(T + T_0)}{R(T^2 + T_0^2)} \right]$$
 (3)

Here, T is temperature in Kelvin and R is ideal gas constant (0.0019872 kcal mol<sup>-1</sup> K<sup>-1</sup>). The fitting parameters A, n, E, and  $T_0$  were listed in Table S7. The temperature-dependent Arrhenius activation energy have been fitted by using eqn (4) as listed in Table 3, which provides the phenomenological characteristics of temperature dependence of the rate constants.

$$E_a(T) = -R \frac{d \ln k}{d (1/T)} \tag{4}$$

The calculated rate constants for  $k_{R1}$  are decreased from 2.64 × 10<sup>-10</sup> cm<sup>3</sup> molecule<sup>-1</sup> s<sup>-1</sup> to 1.83 × 10<sup>-12</sup> cm<sup>3</sup> molecule<sup>-1</sup> s<sup>-1</sup> at 200 - 340 K in Table 2, which shows significant negative temperature dependence. The temperature dependent activation energies are decreased from -3.45 to -5.32 kcal/mol at 200 - 340 K, which provides the evidence for the negative temperature dependence of the rate constants of R1. The rate constant for the reaction of CH<sub>2</sub>OO + HOCH<sub>2</sub>CN is 5.79 × 10<sup>-12</sup> cm<sup>3</sup> molecule<sup>-1</sup> s<sup>-1</sup> at 298 K, which is 24 times larger than the rate constant for the reaction of OH + HOCH<sub>2</sub>CN at 298 K (Marshall and Burkholder, 2024). In particular, we have found that the rate constants of reaction R1 is two magnitude order faster than the reaction of OH + HOCH<sub>2</sub>CN when temperature below 260 K (See Table 2).

The rate constant of the reaction R2 ranges from 3.00 × 10<sup>-14</sup> cm<sup>3</sup> molecule<sup>-1</sup> s<sup>-1</sup> to 2.03 × 10<sup>-15</sup> cm<sup>3</sup> molecule<sup>-1</sup> s<sup>-1</sup> in the temperature range 200 – 340 K, which exhibits weak negative temperature dependence. As a result, we found that *syn*-CH<sub>3</sub>CHOO make a minor contribution for the sink of HOCH<sub>2</sub>CN because the rate constants of *syn*-CH<sub>3</sub>CHOO + HOCH<sub>2</sub>CN are always lower than those of the reaction of OH + HOCH<sub>2</sub>CN at 200 – 340 K. However, the rate constant of reaction of R2 is 2.69 × 10<sup>-15</sup> cm<sup>3</sup> molecule<sup>-1</sup> s<sup>-1</sup> at 298 K, which is two orders of magnitude slower than the rate constant of the reaction of OH + HOCH<sub>2</sub>CN and two orders of magnitude larger than that of the reaction of *syn*-CH<sub>3</sub>CHOO + CH<sub>3</sub>CN, suggesting that the contribution of *syn*-CH<sub>3</sub>CHOO to the sink of HOCH<sub>2</sub>CN is small, yet this reaction is more favourable than that for the reaction of *syn*-CH<sub>3</sub>CHOO + CH<sub>3</sub>CN in the atmosphere (Zhang et al., 2022). Therefore, we do not consider the contribution of *syn*-CH<sub>3</sub>CHOO to HOCH<sub>2</sub>CN in the atmosphere. Furthermore, for this barrierless reaction between small molecules where the torsional degrees of freedom undergo minimal change, the effects of recrossing, quantum tunnelling, and torsional anharmonicity are expected to be negligible This expectation is confirmed by our calculations, which show that the corresponding coefficients are all close to unity, as listed in Tables S5 and S6.

Table 2. The rate constants (cm³ molecule⁻¹ s⁻¹) and activation energies (kcal mol⁻¹) for the CH2OO/syn-CH3CHOO + HOCH2CN reactions at different temperatures.

| T/K | l <sub>ro</sub> , | Fani      | kna               | Eapa             | kou | $k_{\rm DJ}/k_{\rm OM}$ |
|-----|-------------------|-----------|-------------------|------------------|-----|-------------------------|
| 1/1 | $\kappa_{\rm R1}$ | $Ea_{R1}$ | $\kappa_{\rm R2}$ | La <sub>R2</sub> | КОН | $K_{\rm R1}/K_{\rm OH}$ |

| 200 | $2.64 \times 10^{-10}$ | -3.45 | $3.00 \times 10^{-14}$ | -7.65 | $1.06 \times 10^{-13}$ | 2495.80 |
|-----|------------------------|-------|------------------------|-------|------------------------|---------|
| 220 | $1.32 \times 10^{-10}$ | -4.33 | $8.16 \times 10^{-15}$ | -3.54 | $1.25 \times 10^{-13}$ | 1056.27 |
| 240 | $5.26 \times 10^{-11}$ | -4.93 | $5.20 \times 10^{-15}$ | -2.55 | $1.48 \times 10^{-13}$ | 354.83  |
| 250 | $3.39 \times 10^{-11}$ | -5.14 | $4.46 \times 10^{-15}$ | -2.33 | $1.61 \times 10^{-13}$ | 210.30  |
| 260 | $2.25 \times 10^{-11}$ | -5.31 | $3.92 \times 10^{-15}$ | -2.18 | $1.76 \times 10^{-13}$ | 127.83  |
| 270 | $1.52 \times 10^{-11}$ | -5.43 | $3.16 \times 10^{-15}$ | -2.02 | $1.91 \times 10^{-13}$ | 79.59   |
| 280 | $1.05 \times 10^{-12}$ | -5.52 | $3.16 \times 10^{-15}$ | -2.02 | $2.07 \times 10^{-13}$ | 50.90   |
| 298 | $5.79 \times 10^{-12}$ | -5.59 | $2.69 \times 10^{-15}$ | -1.98 | $2.39 \times 10^{-13}$ | 24.26   |
| 300 | $5.44 \times 10^{-12}$ | -5.59 | $2.65 \times 10^{-15}$ | -1.98 | $2.42 \times 10^{-13}$ | 22.45   |
| 320 | $3.04 \times 10^{-12}$ | -5.55 | $2.29 \times 10^{-15}$ | -1.99 | $2.81 \times 10^{-13}$ | 10.84   |
| 340 | $1.83 \times 10^{-12}$ | -5.42 | $2.03 \times 10^{-15}$ | -2.04 | $3.22 \times 10^{-13}$ | 5.68    |

# 3.3 Atmospheric implications

The reaction of OH with HOCH<sub>2</sub>CN has been investigated in previous work (Marshall and Burkholder, 2024).

Therefore, we considered the competition between the R1 reaction and the OH + HOCH<sub>2</sub>CN reaction by comparing with their rate ratios followed by eqn (5),

$$v_1 = \frac{k_{R1}[\text{CH}_2\text{OO}][\text{HOCH}_2\text{CN}]}{k_{\text{OH}}[\text{OH}][\text{HOCH}_2\text{CN}]} = \frac{k_{R1}[\text{CH}_2\text{OO}]}{k_{\text{OH}}[\text{OH}]}$$
 (5)

where  $k_{R1}$  is referred to the rate constants of the reaction of CH<sub>2</sub>OO + HOCH<sub>2</sub>CN,  $k_{OH}$  is rate constants of the reaction of OH with HOCH<sub>2</sub>CN from the literature (Marshall and Burkholder, 2024).

In the atmosphere, Vereecken et al. have evaluated that the concentrations for stabilized Criegee intermediates are in the range between 10<sup>4</sup> and 10<sup>5</sup> molecule cm<sup>-3</sup>, especially in the Amazon rainforest region, where sCls could reach a maximum concentration of 10<sup>5</sup> molecule cm<sup>-3</sup> (Vereecken et al., 2017). Typically, the concentration of OH varies between 10<sup>4</sup> and 10<sup>6</sup> molecules cm<sup>-3</sup> (Ren et al., 2003; Stone et al., 2012; Lelieveld et al., 2016). However, due to the consideration of reactions CH<sub>2</sub>OO with H<sub>2</sub>O and (H<sub>2</sub>O)<sub>2</sub>, the CH<sub>2</sub>OO concentration in the GEOS-Chem model simulations is always an order of magnitude less than the results of Vereecken et al. Therefore, we consider the rate ratios between CH<sub>2</sub>OO + HOCH<sub>2</sub>CN and OH + HOCH<sub>2</sub>CN at different concentrations of CH<sub>2</sub>OO and OH at 200-340 K in Table 3, and further discuss the atmospheric implications based on global atmospheric chemistry model GEOS-Chem.

Table 3. The concentration ratio of CH<sub>2</sub>OO to OH at different heights from different region in GEOS-Chem.

| T/K   | P/mbar                                             | [CH <sub>2</sub> OO] <sup>a</sup>                                                                  | [OH] <sup>a</sup>                                                                                                                            | [CH <sub>2</sub> OO]/[OH] <sup>a</sup>                | $k_{\rm R1}/k_{ m OH}$                                | $\mathbf{v_1}^{b}$                                    |
|-------|----------------------------------------------------|----------------------------------------------------------------------------------------------------|----------------------------------------------------------------------------------------------------------------------------------------------|-------------------------------------------------------|-------------------------------------------------------|-------------------------------------------------------|
|       |                                                    |                                                                                                    | India                                                                                                                                        |                                                       |                                                       |                                                       |
| 290.2 | 1013                                               | 253.44                                                                                             | $1.92 \times 10^{4}$                                                                                                                         | $1.32 \times 10^{-2}$                                 | 33.27                                                 | 0.44                                                  |
| 250.5 | 495.9                                              | 23.10                                                                                              | $1.40 \times 10^{4}$                                                                                                                         | $1.65 \times 10^{-3}$                                 | 203.47                                                | 0.34                                                  |
| 215.6 | 242.8                                              | 20.22                                                                                              | $6.24 \times 10^{3}$                                                                                                                         | $3.24 \times 10^{-3}$                                 | 1262.75                                               | 4.09                                                  |
|       |                                                    | the s                                                                                              | outheast of China                                                                                                                            | ļ                                                     |                                                       |                                                       |
| 290.2 | 1013                                               | 95.21                                                                                              | $1.49 \times 10^{4}$                                                                                                                         | $6.39 \times 10^{-3}$                                 | 33.27                                                 | 0.21                                                  |
| 250.5 | 495.9                                              | 19.22                                                                                              | $1.49 \times 10^{4}$                                                                                                                         | $1.29 \times 10^{-3}$                                 | 203.47                                                | 0.26                                                  |
| 215.6 | 242.8                                              | 11.34                                                                                              | $1.08 \times 10^{4}$                                                                                                                         | $1.05 \times 10^{-3}$                                 | 1262.75                                               | 1.33                                                  |
|       |                                                    | Indone                                                                                             | sian and Malaysi                                                                                                                             | an                                                    |                                                       |                                                       |
| 290.2 | 1013                                               | 150.42                                                                                             | $3.45 \times 10^{4}$                                                                                                                         | $4.36 \times 10^{-3}$                                 | 33.27                                                 | 0.15                                                  |
| 250.5 | 495.9                                              | 79.21                                                                                              | $1.38 \times 10^{4}$                                                                                                                         | $5.74 \times 10^{-3}$                                 | 203.47                                                | 1.17                                                  |
|       | 290.2<br>250.5<br>215.6<br>290.2<br>250.5<br>215.6 | 290.2 1013<br>250.5 495.9<br>215.6 242.8<br>290.2 1013<br>250.5 495.9<br>215.6 242.8<br>290.2 1013 | 290.2 1013 253.44 250.5 495.9 23.10 215.6 242.8 20.22  the s  290.2 1013 95.21 250.5 495.9 19.22 215.6 242.8 11.34  Indone 290.2 1013 150.42 | $\begin{array}{c ccccccccccccccccccccccccccccccccccc$ | $\begin{array}{c ccccccccccccccccccccccccccccccccccc$ | $\begin{array}{c ccccccccccccccccccccccccccccccccccc$ |





The competition between the CH<sub>2</sub>OO + HOCH<sub>2</sub>CN reaction and the OH + HOCH<sub>2</sub>CN reaction is determined by two factors, one is the rate constant ratio and the other is the concentration ratio. The concentration ratio decreases with increasing altitude until two orders of magnitude are observed at 10 km in the Indonesian and Malaysian regions. It is noted that when the altitude increases, the atmospheric temperature is remarkably decreased. The atmospheric temperatures are 250.5-198 K at the altitudes from 5-15 km. The results show that although the concentration ratio of two orders of magnitude, CH<sub>2</sub>OO does make some contribution to the sink of HOCH<sub>2</sub>CN at 1 km. At 10 km, CH<sub>2</sub>OO + HOCH<sub>2</sub>CN can completely dominate over the OH + HOCH<sub>2</sub>CN reaction in India, southeast China, Indonesia, and Malaysian region (See Table 3 and Figure 4). However, CH<sub>2</sub>OO only dominates over the sink of HOCH<sub>2</sub>CN in the Indonesian and Malaysian regions due to the relatively large ratio of rate constants and concentration ratios at low temperatures at 5 km. Using the model data, we find that the sink of CH<sub>2</sub>OO to HOCH<sub>2</sub>CN has strong geographical and sensitivity to altitude (See Fig. 4.).

The reaction products of HOCH<sub>2</sub>CN with OH radicals exhibit significant differences from those formed by the reaction of CH<sub>2</sub>OO with HOCH<sub>2</sub>CN. The main products of the HOCH<sub>2</sub>CN + OH reaction are H<sub>2</sub>O and the HOC(H)CN radical, which subsequently reacts with O<sub>2</sub> to yield HO<sub>2</sub> and formyl cyanide (HC(O)CN) (Marshall and Burkholder, 2024). In contrast, the reaction of HOCH<sub>2</sub>CN with CH<sub>2</sub>OO proceeds through chemical transformation processes, ultimately forming CO and glycolamide. Glycolamide is an amide, which can contribute to the formation of secondary organic aerosols and an important interstellar molecule (Joshi and Lee, 2025; Sanz-Novo et al., 2020; Yao et al., 2016).

<sup>&</sup>lt;sup>a</sup> the data were extracted in the work of Long et al. (Long et al., 2024)

b the product of rate ratio and concentration ratio.

Figure 4: The ratio of CH<sub>2</sub>OO to OH at night from literature.(Long et al., 2024) (a) at 1 km, (b) at 5 km, (c) at 10 km, (d)at 15 km.

# **4 Conclusions**







Wildfires have attracted the attention of researchers due to their impact on aerosols and the ozone layer, which can lead to adverse effects on the environment and human health, HOCH<sub>2</sub>CN is a harmful species present in wildfires. Thus, it is necessary to study its atmospheric chemical processes. The quantitative kinetics for reactions of Criegee intermediates with HOCH<sub>2</sub>CN have been investigated by using specific computational strategies for electronic structure calculations and duallevel strategy for kinetics calculations coupled with atmospheric chemistry transport model analysis. The high-accuracy quantum chemical calculations were performed by using W3X-L//DF-CCSD(T)-F12b/jun-cc-pVDZ-F12 for reaction of R1 close to CCSDT(O)/CBS accuracy and an approximation strategy to reach W3X-L accuracy for R2. Additionally, CCSD(T)-F12a/cc-pVDZ-F12 was used to verified the reliability of DF-CCSD(T)-F12b/jun-cc-pVDZ-F12 for reaction R1. Four mechanisms were found for the reactions of CH2OO and HOCH2CN, with the lowest energy pathway route called carbonoxygen addition coupled carbon-nitrogen addition. We find an unprecedentedly low  $\Delta H_0^{\dagger}$  of -5.61 kcal/mol for the reactions of CH<sub>2</sub>OO with C≡N group of atmospheric species. Simultaneously, we show that the final product in the CH<sub>2</sub>OO + HOCH<sub>2</sub>CN reaction is glycolamide and CO, where glycolamide could contribution to the formation of secondary organic aerosols. The present findings uncover that the post-CCSD(T) is necessary to obtain quantitative  $\Delta H_0^{\ddagger}$  because its contribution is 0.58 kcal/mol. However, we also find that all the factors contain anharmonicity, recrossing and tunneling, torsional anharmonicity effects, which are negligible for obtaining quantitative rate constants. The rate constants for the CH<sub>2</sub>OO with HOCH<sub>2</sub>CN increase from 3.18 × 10<sup>-10</sup> to 2.25 × 10<sup>-11</sup> cm<sup>3</sup> molecule<sup>-1</sup> s<sup>-1</sup>, which is two orders of magnitude higher than the reaction of OH + HOCH<sub>2</sub>CN below 260 K. Therefore, the reaction of CH<sub>2</sub>OO with HOCH<sub>2</sub>CN dominates over the sinks of HOCH<sub>2</sub>CN in southeast China, northern India at 5 km and in the Indonesian and Malaysian regions at 5 and 10 km. This work provides a new insight into the role of Criegee intermediate in the removal of HOCH<sub>2</sub>CN.

**Supplement.** The following information is provided in the Supplement: Standard scale factors and Specific Reaction Scale Factors; The activate enthalpies at 0 K for the CH<sub>2</sub>OO + HOCH<sub>2</sub>CN reaction at different methods; The anharmonicity effect for the enthalpy of activation; The rate constants of the reaction of R1 and R2; The fitting parameters for k<sub>1</sub> and k<sub>2</sub>; The ratio of the rate constants at various temperatures and different concentration; Absolute energies (Hartree) and the Cartesian coordinates (Å) of the optimized geometries; The relative enthalpies at 0 K for the reaction of CH<sub>2</sub>OO + HOCH<sub>2</sub>CN.

**Data availability.** All raw data can be provided by the corresponding authors upon request.

**Author contributions.** Chaolu Xie performed the calculations, analysed and interpretation of data, and wrote the manuscript draft. Shunyu Li performed the calculations. Bo Long designed the project, analysed and interpretation of data, and reviewed and edited the manuscript.


**Competing interests.** The authors declare that they have no conflict of interest.

Acknowledgements. We also thank the Minnesota Supercomputing Institute for computational resources

Financial support. This work was supported in part by the National Natural Science Foundation of China (42120104007 and 41775125,) and by Guizhou Provincial Science and Technology Projects, China (CXTD 2022001 and GCC 2023026).

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
