# Peer review of "Reaction between Criegee intermediates and hydroxyacetonitrile: Reaction mechanisms, kinetics, and atmospheric implications"

_EGUsphere, 2025_

## Author Comment (AC1)

MS No: egusphere-2025-2580

MS type: Research article

Title: Reaction between Criegee intermediates and hydroxyacetonitrile: Reaction mechanisms, kinetics, and atmospheric implications

Author(s): Chaolu Xie, Shunyu Li, Bo Long

Responses in blue.

**CC: 1**

The article describes quantitative reaction kinetics calculations performed for the reaction between Criegee Intermediates and hydroacetonitrile. The calculations are well done, and the reaction in question is atmospherically relevant, as evidenced by the GEOS-Chem modelling, and therefore I gladly recommend the publication of this manuscript in Atmospheric Chemistry and Physics.

However, I have a few minor issues I hope that the authors will address first, mostly but not exclusively relating to presentation and clarity of the manuscript.

Suggestions related to calculations & results:

1.Lines 160-162: "We mainly consider the most feasible mechanism in detail in this work because the enthalpy of activation via TS1 at 0 K is at least 5 kcal/mol lower than those of other reaction pathways by M11-L/MG3S (See Fig. 1). Moreover, the intermediate product M1 formed has a larger enthalpy of activation of –58.33 kcal/mol in Fig.1"

While I agree that the five-ring closure is probably the most competitive reaction, I would also like to point out that you have a barrierless reaction with negative T-dependence 1-2 orders of magnitudes slower than the collision limit. This is evidently a case where the entropy also matters, and this raises the possibility of the H-shift reaction (TS1a + TS1b) being possible as a minor channel, due to it likely being less entropically restricted than the ring formation. I suggest you add a table or a plot to the supplementary of the Gibbs free energies of TS1, TS1a, and TS1b over the considered T range to check if if might be relevant.

**Author response:** Thanks for your comments. We have added a Gibbs free energy profile for the $CH_2OO$ + $HOCH_2CN$ reaction in Figure S2, which shows that the five-ring closure reaction pathway remains the most favorable pathway due to the relatively large Gibbs free energy of the hydrogen transfer reaction. The Gibbs free energy diagram is shown below. We have added the discussion in the revised article. "In addition, the calculated Gibbs free energy barriers at 298 K also show the five-ring closure reaction pathway via TS1 is the lowest in the $CH_2OO$ + $HOCH_2CN$ reaction (See Figure S2); this reveals that entropy has a negligible effect on reaction mechanism." In line 163-165 on page 8.

[Figure]

Figure S2. The relative Gibbs free energy at 298 K for the reaction of CH₂OO + HOCH₂CN. Values are given for all species as calculated by M11-L/MG3S. in small parentheses, values are given for the transition state TS1 calculated by W3X-L//DF-CCSD(T)-F12b/jun-cc-pVDZ.

2.The Supplement is missing the xyz geometries for all species in Figures 3 & S1. Please add them.

**Author response:** Thanks for your comments. We have added the xyz coordinates for all species in Table S8.

3.And here's the big one:

There's a combined experimental/computational article on Criegee + C≡N reactions published quite soon after yours (https://doi.org/10.1021/acs.jpca.2c07073. Full disclosure: I am one of the co-authors on this article), that found quite different decomposition channels for the 5-membered ring product. Our decomposition channels have higher 0 K energies than yours, but they are presumably more entropically favourable, which especially matters for the product distribution when the intermediate product forms with 51-58 kcal/mol worth of excess energy. What's more, now that I got aware of your calculations on the topic, I tried to do a saddle point search on your M2-TSb transition state using wB97X-D3 based on the figure, without success. This makes me question whether this N-C-O ring closure saddle point may actually be an artefact produced by M11-L. I would be very interested in seeing calculations of this barrier using some other levels of theory as well, and preferably also a comparison to our CH2O ejection saddle point. Please also report entropies in addition to 0 K energies, since all of these saddle point are

well below the energy of the free reactants, and since we have shown that only a small fraction of the five-membered ring stabilized for $CH_2OO$ + Acetonitrile.

**Author response:** Thanks for your comments. In the revised article, we reproduced the transition state (TSD) reported in that paper (Franzon et al., 2023) using $\omega$B97XD/MG3S and $\omega$B97XD/aug-cc-pVTZ methods. The calculated output files are provided in the supplementary material as Output.zip. Then, we did intrinsic reaction coordinate (IRC) calculations as shown in Figure A1. These calculated results indicate that TSD generates a three-membered ring structure similar to M3 in our paper, rather than the 1,2,4-ring-$CH_3$ mentioned in that paper. Moreover, the reaction pathway via TSD ultimately leads to the formation of $CH_3NCO$ and HCHO by $\omega$B97XD/MG3S. We found a transition state (M3-TSc1) similar to TSD in the $CH_2OO$ + $HOCH_2CN$ reaction by $\omega$B97XD/MG3S and M11-L/MG3S and did the IRC calculation. The calculated results reveal the similar reaction mechanism via TSD in the $CH_2OO$ + $CH_3CN$ reaction, indicating that the M3 decomposes into $HOCH_2NCO$ and HCHO via the transition state M3-TSc1. However, we found that the enthalpy of activation at 0 K for M3-TSc1 is 11.97 kcal/mol higher than M3-TSc at the M11-L/MG3S level.

We also used four different methods to calculate this transition state (M2-TSb) by MN12-L/MG3S, MN15-L/MG3S, M06-2X/MG3S, and $\omega$B97XD/MG3S, which is provided in the supplementary material as Output.zip. The four methods can give a ring-closed transition state (M2-TSb), confirming that this is not an artifact of the M11-L/MG3S method.

We have added some comments in the revised article. "Then, the three-member ring intermediate M3 then undergoes two different reaction routes. One is open-ring coupled hydrogen shift to form M4 via the transition state M3-TSc. The other is analogous to that proposed by Franzon et al. (Franzon et al., 2023), which is an open-ring coupled bond breaking to form HCHO and $HOCH_2NCO$ via the transition state M3-TSc1. However, the enthalpy of activation at 0 K for M3-TSc is 11.97 kcal/mol lower than that of M3-TSc1. Therefore, M3-TSc is the dominant reaction pathway for the unimolecular reaction of M3. Moreover, IRC calculations also show that M3-TSc connect well with M3 as described in Figure S3." in line 206-211 on page 10. And "The intrinsic reaction coordinate (IRC) was performed by M11-L/MG3S, and the results were depicted the results in Figure S3." in line 95-96 on page 4.

[Figure]

Figure A1. The IRC calculated results by $\omega$B97XD/MG3S.

Discussion-related comments:

4.In general: You use abbreviations of "Higher level" and "lower level", but you write out "enthalpy of activation at 0 K" a total of 11 times throughout the manuscript. How about abbreviating it as H_0?

**Author response:** Thanks for your comments. We have corrected the term "enthalpy of activation at 0 K" to "$\Delta H_0^{\ddagger}$" and "reaction enthalpy at 0 K" to "$\Delta H_0$" in the revised version. We have added the comment. "For simplicity, the activation enthalpy at 0 K is defined as the difference between the transition state and the reactants, abbreviated as $\Delta H_0^{\ddagger}$. The difference between the products and the reactants is defined the reaction enthalpy at 0 K, abbreviated as $\Delta H_0$." in line 68-70 on page 3.

5.In the abstract and on line 53, you say that are "designing" a computational strategy, but it seems to me that you are simply using a previously established computational strategy that you have already utilized in several published papers. Please rephrase.

**Author response:** Thanks for your comments. We have indeed used this method before, and we have rephrased. "We used the approximation strategy mentioned previously (Sun et al., 2023) to obtain the accuracy close to W3X-L in equation (1) for *syn*-$CH_3CHOO + HOCH_2CN$ to reduce computational costs." in line 87-88 on page 4. and "we employ computational strategies approaching CCSDT(Q)/CBS accuracy, combined with a dual-level strategy, to unravel the key factors governing the reaction kinetics." in line 10-11 on page 1.

6.Lines 21-22: "hydroxyacetonitrile ($HOCH_2CN$) has been recently identified as a C2H3NO isomer by

using I− chemical ionization mass spectrometry (I-CIMS) instrument detection."

You do not need an instrument to identify HOCH2CN as an isomer of C2H3NO. It's in the chemical formula.

**Author response:** Thanks for your comments. We have corrected the presentation. "Hydroxyacetonitrile ($HOCH_2CN$), a reactive nitrogen-containing compound, has recently been identified as a $C_2H_3NO$ isomer. Earlier field measurements had attributed the $C_2H_3NO$ signal to methyl isocyanate ($CH_3NCO$) when using chemical ionization mass spectrometry (CIMS) (Priestley et al., 2018; Mattila et al., 2020; Wang et al., 2022a), as CIMS is insensitive to the detection of isomers, and thus cannot differentiate between the isomers of CH3NCO and HOCH2CN. However, very recent detection with I− chemical ionization mass spectrometry (I-CIMS) identified the $C_2H_3NO$ signal as $HOCH_2CN$ (Finewax et al., 2024)." in line 21-24 on page 2.

7.Lines 21-28: "However, previously several field studies had misattributed the $C_2H_3NO$ signal to methyl isocyanate ($CH_3NCO$) by using CIMS. CIMS is insensitive to the detection of isomers, and thus cannot differentiate between the isomers of CH3NCO and HOCH2CN. CH3NCO had been detected in chemicals released from biomass burning, such as wildfires and agricultural fires, as well as in bleach cleaning environments. Therefore, the atmospheric sources of CH3NCO from previous investigations are actually the sources of HOCH2CN."

I am unable to follow the argumentation here. If I may simplyfy, it seems that it says "HOCH2CN is an isomer of CH3NCO. Therefore previous observations of CH3NCO must have been HOCH2CN in reality." Slightly more evidence is needed to claim this.

**Author response:** Thanks for your comments. We have rephrased this and added the corresponding references. "Hydroxyacetonitrile ($HOCH_2CN$), a reactive nitrogen-containing compound, has recently been identified as a $C_2H_3NO$ isomer. Earlier field measurements had attributed the $C_2H_3NO$ signal to methyl isocyanate ($CH_3NCO$) when using chemical ionization mass spectrometry (CIMS) (Priestley et al., 2018; Mattila et al., 2020; Wang et al., 2022a), as CIMS is insensitive to the detection of isomers, and thus cannot differentiate between the isomers of $CH_3NCO$ and $HOCH_2CN$. However, very recent detection with I− chemical ionization mass spectrometry (I-CIMS) identified the $C_2H_3NO$ signal as $HOCH_2CN$ (Finewax et al., 2024)." and "In other words, the $CH_3NCO$ detected in the atmosphere is essentially $HOCH_2CN$. Therefore, the atmospheric sources of $CH_3NCO$ from previous investigations are actually the sources of $HOCH_2CN$. Consequently, $HOCH_2CN$ is emitted in chemicals released from biomass burning, such as wildfires and agricultural fires, as well as in bleach cleaning environments(Mattila et al., 2020; Priestley et al., 2018; Wang et al., 2022; Koss et al., 2018; Papanastasiou et al., 2020)." in line 27-30 on page 2.

8.Lines 41-42: "Criegee intermediate is considered to be a key intermediate due to its effect on the atmosphere (Chhantyal-Pun et al., 2020b)."

This sentence adds nothing that was not already said in the previous sentence. You can remove it easily.

**Author response:** Thanks for your comments. We have removed this sentence.

9.Lines 49-51: "While reaction kinetics of Criegee intermediates are prerequisite for elucidating their chemical processes and finding new sink pathways in the atmosphere, their kinetics are very limited and even unknown."

It's unclear what is meant by this sentence, but it seems that it is trying to say that our knowledge of the

reaction kinetics of Criegee intermediates is very limited. This is nonsense, as there are plenty of articles on the topic, including all those that were cited in this paragraph. Please rephase to make it clearer what you mean.

**Author response:** Thanks for your comments. We have corrected the sentence in the revised version. "While reaction kinetics of Criegee intermediates with $HOCH_2CN$ are prerequisite for elucidating its chemical processes and finding new sink pathways in the atmosphere, its kinetics are unknown." in line 51-52 on page 3.

10.Line 55: "the quantitative enthalpy of activation at 0 K acted as high level is used to calculate the rate constant by using conventional transition state theory without tunnelling"

This sentence is nonsensical. It looks like you have changed your mind about what the subject should be mid-sentence.

**Author response:** Thanks for your comments. We have corrected this sentence as "In our dual-level strategy, W3X-L//DF-CCSD(T)-F12b/jun-cc-pVDZ is used to obtain conventional transition state theory rate constants, while validated DFT methods capture recrossing and tunnelling effects through CVT with small-curvature tunnelling." in line 56-58 on page 3.

11.On Line 61, a citation in needed for GEOS-Chem.

**Author response:** Thanks for your comments. We have already added the reference in line 61 on page 3.

12.Line 68-69: "The basic requirement in kinetics calculations is the quantitative enthalpy of activation that is determined…"

You are making it sound like reaction kinetics is all about determining the entrahlpy of activation, when the statistical mechanics description of gas-phase chemical reactions also crucially depends on entropies in thermalized conditions at non-zero temperatures, as well as excess energy, collisional stabilisation, and fall-off effects in non-thermalized conditions. And then there are of course tunneling effects. Having a quantitatively accurate enthalpy of activation does not matter if all of these physical effects are modelled poorly, and I presume you know this, considering how much effort you have put into getting them right.

**Author response:** Thanks for your comments. We fully agree that the rate constant is governed not only by the enthalpy of activation but also by entropies in thermalized conditions at non-zero temperatures, as well as excess energy, collisional stabilization, and fall-off effects in non-thermalized conditions. However, quantitative activation enthalpy is a prerequisite for obtained quantitative kinetics paraments, because a change of 1 kcal in activation enthalpy can change its rate constant by 5.4 times according to the Arrhenius equation. We have revised this sentence to "Gas-phase chemical reactions also crucially depend on entropies in thermalized conditions at non-zero temperatures, as well as excess energy, collisional stabilization, and fall-off effects in non-thermalized conditions. However, accurate determination of $\Delta H_0^{\ddagger}$ remains essential for calculating quantitative kinetics." in line 71-74 on page 3.

13.Line 109: "ktight was calculated by using a dual-level strategy discussed below"

The higher and lower levels of theory were already described previously in the paper. Therefore, you are causing a lot of confusion for the reader by referring to "a dual-level strategy discussed below" here. It

made me think "Wait, wasn't the dual-level strategy already described? Is this a different dual-level strategy? Was I supposed to read this before the previous page?" etc. Please rephrase.

**Author response:** Thanks for your comments. The dual-level strategy mentioned in the work is a strategy for calculating rate constants. This strategy refers to the combination of the theory of conventional transition states on the HL with the theory of canonical variational transition state on the LL and takes into account the small curvature tunnelling effect. We have modified the sentence to "where $k_{tight}$ was calculated by using a dual-level strategy described in detail below". in line 115 on page 5.

14. Line 112-113: "the reaction coordinate s is obtained by defining the distance between one pivot point on one reactant and the other pivot point on the other"

How were these pivot points determined? Was it the centre of mass of each molecule or something else?

**Author response:** Thanks for your comments. The pivot point is located in a vector at a distance d from the center of mass (COM) of the reactants. We have modified the sentence to "In VRC-TST calculation, the reaction coordinate $s$ is obtained by defining the distance between one pivot point on one reactant and the other pivot point on the other, and the dividing surface is defined by the pivot point connected to each reactant. The pivot point is located in a vector at a distance d from the center of mass (COM) of the reactants, which is chosen to minimize the reaction rate. The vector connecting the pivot point to the center of mass of reactant and is perpendicular to the plane of the reactant." in line 117-121 on page 5.

15. Line 129: "More details were provided in Tables S4 and S5." (Also on Line 226)

(1). I think you mean "Tables S5 and S6".

**Author response:** Thanks for your comments. We have modified the sentence to "The calculation each component present in equation (2) were provided in Tables S5 and S6." in line 137 on page 5 and "The rate constants for the reaction of R1 and R2 have been calculated and listed in Table 2." in line 237 on page 12.

(2). What you have in these tables can not exactly be described as "more details". I would describe it as a factorization of the rate coefficient into each component present in equation (2).

**Author response:** Thanks for your comments. We have modified the sentence to "The calculation each component present in equation (2) were provided in Tables S5 and S6." in line 137 on page 6.

(3). Table S2 requires an explanation of what the two lambdas represent, and how the accurate anharmonic ZPEs were calculated to determine the reaction-specific scaling factors.

**Author response:** Thanks for your comments. We have added explanation for the factors in Table S2.

16. Line 205: "The calculated enthalpy of activation at 0 K of P1 is – 205 86.55 kcal/mol, indicating the unimolecular isomerization is –5.61 thermodynamically driven"

Please rephrase this. It is unclear what number you are referring to with "-5.61 themodynamically driven.", and what you mean by it.

**Author response:** Thanks for your comments. We have modified the sentence to "The calculated enthalpy of activation at 0 K of P1 is –86.55 kcal/mol, indicating the unimolecular isomerization is thermodynamically driven." in line 217-218 on page 10.

17. Line 222-223: "Therefore, the enthalpy of activation at 0 K for every reaction can only be quantitively

obtained by specific calculations."

Here it is again unclear what you mean. Do you mean that the barriers must be calculated separately for each reaction instead of estimated based on literature data? I'm sure everyone in the field agrees that it is better to do so in principle, but this does not invalidate the staregy of making rate estimates whenever possible. Please either rephrase, or possibly remove the sentence completely.

**Author response:** Thanks for your comments. We have removed the sentence in the revised version.

18.Lines 230-231: "The temperature-dependent Arrhenius activation energies also have been fitted by using eqn (4) as listed in Table 3."

Typically, Ea is a constant in the Arrhenius equation. While this is already impled, you should add that you are fitting to the Arrhenius-like equation $k = A \exp(-Ea(T)/RT)$ instead of $k = A(T) \exp(-Ea/RT)$, for example.

**Author response:** Thanks for your comments. We have modified the sentence to "The temperature-dependent Arrhenius activation energy have been fitted by using eqn (4) as listed in Table 3, which provides the phenomenological characteristics of temperature dependence of the rate constants." in line 239-240 on page 12.

19.Line 235: "which provides the evidence for the negative temperature dependence of the rate constants of R1."

The activation energy does not provide evidence for anything. Your quantitative rate calculations provide the evidence, and the T-dependence of Ea is just those results fit to a simple function.

**Author response:** Thanks for your comments. The temperature-dependent Arrhenius activation energy can be expressed as the slope in the local Arrhenius fit. If Ea is negative, this indirectly proves that its rate constant has negative temperature dependence.

20.Lines 236-244: There are several parts here where you forget to say "at 298 K" when citing a specific value of k.

**Author response:** Thanks for your comments. We have corrected this problem.

21.Table 2: The footnotes a & b are unnecessary if you use the subscripts kR1 and kR2 for the reaction rates instead of k1 and k2.

**Author response:** Thanks for your comments. We have replaced all $k_1$ with $k_{R1}$ and all $k_2$ with $k_{R2}$.

22.Lines 248-250: "Additionally, we have found that recrossing, tunneling transmission, and torsional anharmonicity effects for the reaction of R1 and R2 can be negligible because they are close to unit, as listed in Table S5 and S6."

There is nothing wrong with this sentence, but I want to point out that all of this is to be expected for a barrierless reaction for two small molecules where the torsional modes of freedom (the O-H, C-C(OH) and for syn-Criegee the -CH3 bond rotations) undergo no change during the reaction.

**Author response:** Thanks for your comments. We fully agree your comment. We have added comment in the revised version. "Furthermore, for this barrierless reaction between small molecules where the torsional degrees of freedom undergo minimal change, the effects of recrossing, quantum tunneling, and torsional anharmonicity are expected to be negligible This anticipation is confirmed by our calculations, which show that the corresponding coefficients are all close to unity, as listed in Tables S5 and S6."

23.Line 262-264: "In the atmosphere, Vereecken et al. have evaluated that the concentrations for stabilized Criegee intermediates are in the range between 104 and 105 molecule cm−3, especially in the Amazon rainforest region, where sCls could reach a maximum concentration of 105 molecule cm−3 (Novelli et al., 2017)."

(1).Vereecken et al. is missing a citation.

**Author response:** Thanks for your comments. We have added the citation about Vereecken et al.

(2). The cited Novelli et al. article discusses measurement of [Criegee] in boreal forest environments, not rainforest environments.

**Author response:** Thanks for your comments. There is a problem with this citation. We have checked the references in the full text and corrected them.

(3). The wording leaves it unclear if which Criegee concentration you trust more, Vereecken's estimate or GEOS-Chem's model. Please also motivate your judgement.

**Author response:** Thanks for your comments. Due to the uncertainty of the source of $CH_2OO$ and its rapid reaction with $H_2O$ and $(H_2O)_2$ in GEOS-Chem, the concentration is lower than that observation in the field.

24.Typos and grammatical errors:

In the Abstract:

"species containing C≡N group" → "containing a C≡N group."

"that the rate constants of" → "that the rate constant of"

"at below 260 K." → "below 260 K."

Line 55: "dual delve" → dual-level

Line 60: "the corresponding OH radical" → "the corresponding OH radical reactions"

Line 87: "The reliable density functional method was chosen" → "A reliable density functional method was chosen".

I hope you were not trying to imply that M11-L is the only reliable density functional method. :)

Line 116: "performed by M06-CR/MG3S " → "performed using M06-CR/MG3S"

Line 139: "there are three different…" → "there are three different functional groups in HOCH2CN:"

Line 141: "Three of the is" → "Three of them are"

Lines 147-148: The words "addition of" are entirely superfluous in the sentence describing the two addition reactions, due to the use of "is added to" later.

Line 249: "can be negligible" → "are negligible"

**Author response:** Thanks for your comments. We have corrected these errors

Franzon, L., Peltola, J., Valiev, R., Vuorio, N., Kurtén, T., and Eskola, A.: An Experimental and Master Equation Investigation of Kinetics of the $CH_2OO$ + RCN Reactions (R = H, $CH_3$, $C_2H_5$) and Their Atmospheric Relevance, The Journal of Physical Chemistry A, 127, 477-488, 10.1021/acs.jpca.2c07073, 2023.

Koss, A. R., Sekimoto, K., Gilman, J. B., Selimovic, V., Coggon, M. M., Zarzana, K. J., Yuan, B., Lerner, B. M., Brown, S. S., Jimenez, J. L., Krechmer, J., Roberts, J. M., Warneke, C., Yokelson, R. J., and de Gouw, J.: Non-methane organic gas emissions from biomass burning: identification, quantification, and emission factors from PTR-ToF during the FIREX 2016 laboratory experiment, Atmos. Chem. Phys., 18, 3299-3319, 10.5194/acp-18-3299-2018, 2018.

Mattila, J. M., Arata, C., Wang, C., Katz, E. F., Abeleira, A., Zhou, Y., Zhou, S., Goldstein, A. H., Abbatt, J. P. D., DeCarlo, P. F., and Farmer, D. K.: Dark Chemistry during Bleach Cleaning Enhances Oxidation of Organics and Secondary Organic Aerosol Production Indoors, Environmental Science & Technology Letters, 7, 795-801, 10.1021/acs.estlett.0c00573, 2020.

Papanastasiou, D. K., Bernard, F., and Burkholder, J. B.: Atmospheric Fate of Methyl Isocyanate, $CH_3NCO$: OH and Cl Reaction Kinetics and Identification of Formyl Isocyanate, HC(O)NCO, ACS Earth and Space Chemistry, 4, 1626-1637, 10.1021/acsearthspacechem.0c00157, 2020.

Priestley, M., Le Breton, M., Bannan, T. J., Leather, K. E., Bacak, A., Reyes-Villegas, E., De Vocht, F., Shallcross, B. M. A., Brazier, T., Anwar Khan, M., Allan, J., Shallcross, D. E., Coe, H., and Percival, C. J.: Observations of Isocyanate, Amide, Nitrate, and Nitro Compounds From an Anthropogenic Biomass Burning Event Using a ToF-CIMS, Journal of Geophysical Research: Atmospheres, 123, 7687-7704, https://doi.org/10.1002/2017JD027316, 2018.

Wang, C., Mattila, J. M., Farmer, D. K., Arata, C., Goldstein, A. H., and Abbatt, J. P. D.: Behavior of Isocyanic Acid and Other Nitrogen-Containing Volatile Organic Compounds in The Indoor Environment, Environmental Science & Technology, 56, 7598-7607, 10.1021/acs.est.1c08182, 2022.

---

## Author Comment (AC3)

MS No: egusphere-2025-2580
MS type: Research article
Title: Reaction between Criegee intermediates and hydroxyacetonitrile: Reaction mechanisms, kinetics, and atmospheric implications
Author(s): Chaolu Xie, Shunyu Li, Bo Long

Responses in blue.

**CC: 2**

This manuscript describes new quantum chemical and computational kinetics calculations that show that reactions of carbonyl oxides are likely to play a significant role in the atmospheric transformations of hydroxyacetonitrile. The calculations are of high reliability and the conclusions are of substantial interest for understanding atmospheric reactions of wildfire products. I recommend publication of the manuscript and have some suggestions for clarification.

1. First, the basis for assessing that hydroxyacetonitrile is in fact a significant atmospheric product in wildfire burning is half described — either the authors should simply refer to the publications that treat the analytical chemistry behind the assignment and its revision (which are lucidly described in the referenced reports) or they should go into more detail here. I would lean towards the first option, but as it is in the current manuscript the story is very confusing.

**Author response:** Thanks for your comments. What we mean here is that the $CH_3NCO$ detected previously is actually $HOCH_2CN$ in the atmosphere by the work of Finewax et al, so the source of $CH_3NCO$ emissions in the atmosphere is actually the source of $HOCH_2CN$. We have modified the sentence to "Hydroxyacetonitrile ($HOCH_2CN$), a reactive nitrogen-containing compound, has recently been identified as a $C_2H_3NO$ isomer. Earlier field measurements had attributed the $C_2H_3NO$ signal to methyl isocyanate ($CH_3NCO$) when using chemical ionization mass spectrometry (CIMS) (Priestley et al., 2018; Mattila et al., 2020; Wang et al., 2022) , as CIMS is insensitive to the detection of isomers, and thus cannot differentiate between the isomers of $CH_3NCO$ and HOCH2CN. However, very recent detection with $I^-$ chemical ionization mass spectrometry (I-CIMS) identified the $C_2H_3NO$ signal as $HOCH_2CN$ (Finewax et al., 2024)." and "In other words, the $CH_3NCO$ detected in the atmosphere is essentially $HOCH_2CN$. Therefore, the atmospheric sources of $CH_3NCO$ from previous investigations are actually the sources of $HOCH_2CN$. Consequently, $HOCH_2CN$ is emitted in chemicals released from biomass burning, such as wildfires and agricultural fires, as well as in bleach cleaning environments (Mattila et al., 2020; Priestley et al., 2018; Wang et al., 2022a; Koss et al., 2018; Papanastasiou et al., 2020)." in line 27-30 on page 2.

2. Second, the manuscript treats simply the control of atmospheric removal of hydroxyacetonitrile, but the implications of this reaction for atmospheric chemistry depend also on the fate of the products and on the (still highly uncertain) tropospheric concentration of carbonyl oxides. The present calculations show a sequence of energetically accessible unimolecular transformations (not really "decomposition") of the initial product. I am wondering if the authors could describe for completeness the lowest bimolecular channels of the reaction, and whether the fate of any of the

proposed isomeric products would be expected to have different consequences for the atmosphere or environment than the others.

**Author response:** Thanks for your comments. The large exothermic, unimolecular conversion of the intermediate M1 formed from $CH_2OO$ + $HOCH_2CN$ is dominant for the formation of the final product, where the rate-determining step is TS1, which similar to the alkene-ozone reaction (Nguyen et al., 2015). We have added the comment in the revised version "Furthermore, the rate-determining step for the formation of the final product from the $CH_2OO$ + $HOCH_2CN$ reaction has been identified as the initial step, which is similar to the reaction of alkenes with ozone(Nguyen et al., 2015)." In line 226-227 on page 10.

3. Also, given that the reaction with OH would be competitive in many environments, would the products of that reaction have different implications than the products of this reaction? I understand that this stretches the boundaries of the work, and perhaps the authors will consider it out of scope, but in an atmospheric chemistry journal I think a bit of additional context in the conclusion section would be fitting.

**Author response:** Thanks for your comments. We have done some additional calculations. As a result, the final products of the reaction between $HOCH_2CN$ and $CH_2OO$ differ from those of the reaction with OH. The dominant products of the reaction between $HOCH_2CN$ and OH are $H_2O$ and HOC(H)CN, which further reacts with $O_2$ to ultimately yield hydroperoxyl radical ($HO_2$) and formyl cyanide (HC(O)CN). However, the reaction of $HOCH_2CN$ with $CH_2OH$ undergoes multiple chemical processes to form HC(O)NHC(O)CH$_2$OH, which finally leads to formation of carbon monoxide and glycolamide. These products are potential toxic air pollutants and significant contributors to atmospheric organic aerosols. We have added some comments in the revised version. "Then, the three-member ring intermediate M3 then undergoes two different reaction routes. One is open-ring coupled hydrogen shift to form M4 via the transition state M3-TSc. The other is analogous to that proposed by Franzon et al. (Franzon et al., 2023), which is an open-ring coupled bond breaking to form HCHO and HOCH$_2$NCO via the transition state M3-TSc1. However, $\Delta H_0^{\ddagger}$ for M3-TS3c is 11.97 kcal/mol lower than that of M3-TSc1. Therefore, M3-TSc is the dominant reaction pathway for the unimolecular reaction of M3. Moreover, IRC calculations also show that M3-TSc connects well with M3 as described in Figure S3. Subsequently, the H atom of the intermediate OH on intermediate M4 is transferred to the N atom to yield the intermediate species M5. Then, the process was depicted in Fig. 3. The calculated enthalpy of reaction at 0 K of M5 is –86.55 kcal/mol, indicating the unimolecular isomerization is thermodynamically driven. Intermediate M5 undergoes unimolecular isomerization via two different pathways. In the first pathway, an intramolecular hydrogen transfer from the aldehyde group to the central carbonyl oxygen is followed by C–N bond cleavage, yielding CO and intermediate M6. Then, hydrogen shift of OH in M6 to NH group leads to the formation of glycolamide. Alternatively, a second pathway involves hydrogen migration from the aldehyde group to the central carbon atom, accompanied by C–N bond rupture, producing HNCO and glycolaldehyde. The formation of carbon monoxide proceeds with a significantly lower activation enthalpy (−54.19 kcal/mol) compared to that for glycolaldehyde (−32.32 kcal/mol), indicating that the CO-forming channel is kinetically favored." in line 212-226 on page 10. "The reaction products of $HOCH_2CN$ with OH radicals exhibit significant differences from those formed by the reaction of $CH_2OO$ with $HOCH_2CN$. The main

products of the HOCH$_2$CN + OH reaction are H$_2$O and the HOC(H)CN radical, which subsequently reacts with O$_2$ to yield HO$_2$ and formyl cyanide (HC(O)CN) (Marshall and Burkholder, 2024). In contrast, the reaction of HOCH$_2$CN with CH$_2$OO proceeds through chemical transformation processes, ultimately forming CO and glycolamide. Glycolamide is an amide, which can contribute to the formation of secondary organic aerosols and an important interstellar molecule (Joshi and Lee, 2025; Sanz-Novo et al., 2020; Yao et al., 2016)." in line 299-304 on page 14. "Simultaneously, we show that the final product in the CH$_2$OO + HOCH$_2$CN reaction is glycolamide and CO, where glycolamide could contribution to the formation of secondary organic aerosols." in line 323-324 on page 15.

**Reference**

Finewax, Z., Chattopadhyay, A., Neuman, J. A., Roberts, J. M., and Burkholder, J. B.: Calibration of hydroxyacetonitrile (HOCH$_2$CN) and methyl isocyanate (CH$_3$NCO) isomers using I− chemical ionization mass spectrometry (CIMS), Atmos. Meas. Tech., 17, 6865-6873, 10.5194/amt-17-6865-2024, 2024.

Franzon, L., Peltola, J., Valiev, R., Vuorio, N., Kurtén, T., and Eskola, A.: An Experimental and Master Equation Investigation of Kinetics of the CH$_2$OO + RCN Reactions (R = H, CH$_3$, C$_2$H$_5$) and Their Atmospheric Relevance, The Journal of Physical Chemistry A, 127, 477-488, 10.1021/acs.jpca.2c07073, 2023.

Joshi, P. R. and Lee, Y.-P.: Identification of the HO•CHC(O)NH2 Radical Intermediate in the Reaction of H + Glycolamide in Solid Para-Hydrogen and Its Implication to the Interstellar Formation of Higher-Order Amides and Polypeptides, ACS Earth and Space Chemistry, 9, 769-781, 10.1021/acsearthspacechem.4c00409, 2025.

Marshall, P. and Burkholder, J. B.: Kinetics and Thermochemistry of Hydroxyacetonitrile (HOCH$_2$CN) and Its Reaction with Hydroxyl Radical, ACS Earth and Space Chemistry, 8, 1933-1941, 10.1021/acsearthspacechem.4c00176, 2024.

Mattila, J. M., Arata, C., Wang, C., Katz, E. F., Abeleira, A., Zhou, Y., Zhou, S., Goldstein, A. H., Abbatt, J. P. D., DeCarlo, P. F., and Farmer, D. K.: Dark Chemistry during Bleach Cleaning Enhances Oxidation of Organics and Secondary Organic Aerosol Production Indoors, Environmental Science & Technology Letters, 7, 795-801, 10.1021/acs.estlett.0c00573, 2020.

Nguyen, T. L., Lee, H., Matthews, D. A., McCarthy, M. C., and Stanton, J. F.: Stabilization of the Simplest Criegee Intermediate from the Reaction between Ozone and Ethylene: A High-Level Quantum Chemical and Kinetic Analysis of Ozonolysis, The Journal of Physical Chemistry A, 119, 5524-5533, 10.1021/acs.jpca.5b02088, 2015.

Priestley, M., Le Breton, M., Bannan, T. J., Leather, K. E., Bacak, A., Reyes-Villegas, E., De Vocht, F., Shallcross, B. M. A., Brazier, T., Anwar Khan, M., Allan, J., Shallcross, D. E., Coe, H., and Percival, C. J.: Observations of Isocyanate, Amide, Nitrate, and Nitro Compounds From an Anthropogenic Biomass Burning Event Using a ToF-CIMS, Journal of Geophysical Research: Atmospheres, 123, 7687-7704, https://doi.org/10.1002/2017JD027316, 2018.

Sanz-Novo, M., Belloche, A., Alonso, J. L., Kolesniková, L., Garrod, R. T., Mata, S., Müller, H. S. P., Menten, K. M., and Gong, Y.: Interstellar glycolamide: A comprehensive rotational study and an astronomical search in Sgr B2(N)⋆, A&A, 639, 2020.

Wang, C., Mattila, J. M., Farmer, D. K., Arata, C., Goldstein, A. H., and Abbatt, J. P. D.: Behavior of Isocyanic Acid and Other Nitrogen-Containing Volatile Organic Compounds in The Indoor Environment, Environmental Science & Technology, 56, 7598-7607, 10.1021/acs.est.1c08182, 2022.

Yao, L., Wang, M. Y., Wang, X. K., Liu, Y. J., Chen, H. F., Zheng, J., Nie, W., Ding, A. J., Geng, F. H., Wang, D. F., Chen, J. M., Worsnop, D. R., and Wang, L.: Detection of atmospheric gaseous amines and amides by a high-resolution time-of-flight chemical ionization mass spectrometer with protonated ethanol reagent ions, Atmos. Chem. Phys., 16, 14527-14543, 10.5194/acp-16-14527-2016, 2016.